# Semi-Supervised Contrastive Learning for Deep Regression with Ordinal Rankings from Spectral Seriation

**Weihang Dai[1], Yao Du[1], Hanru Bai[3], Kwang-Ting Cheng[1], Xiaomeng Li[1,2]***

[1]The Hong Kong University of Science and Technology
[2]HKUST Shenzhen-Hong Kong Collaborative Innovation Research Institute, Futian, Shenzhen
[3]Fudan University
eexmli@ust.hk

## Abstract

Contrastive learning methods can be applied to deep regression by enforcing label distance relationships in feature space. However, these methods are limited to labeled data only unlike for classification, where unlabeled data can be used for contrastive pretraining. In this work, we extend contrastive regression methods to allow unlabeled data to be used in a semi-supervised setting, thereby reducing the reliance on manual annotations. We observe that the feature similarity matrix between unlabeled samples still reflect inter-sample relationships, and that an accurate ordinal relationship can be recovered through spectral seriation algorithms if the level of error is within certain bounds. By using the recovered ordinal relationship for contrastive learning on unlabeled samples, we can allow more data to be used for feature representation learning, thereby achieve more robust results. The ordinal rankings can also be used to supervise predictions on unlabeled samples, which can serve as an additional training signal. We provide theoretical guarantees and empirical support through experiments on different datasets, demonstrating that our method can surpass existing state-of-the-art semi-supervised deep regression methods. To the best of our knowledge, this work is the first to explore using unlabeled data to perform contrastive learning for regression. Code is available at `https://github.com/xmed-lab/CLSS`.

## 1   Introduction

Contrastive learning is an effective technique for improving feature representations in deep neural networks (DNNs) [3, 4, 19, 33, 10]. These methods involve identifying positive and negative sample pairs based on feature similarity, where positive pairs are usually defined as augmented samples from the same input or samples from the same class. Early works such as SimCLR [3], MoCo [4], and SupCon [19], showed that pretraining DNNs through contrastive learning can lead to state-of-the-art classification results due to improved feature representations. These techniques have since been extended to segmentation [33, 43], video recognition [10], multi-modality learning [28], as well as deep regression [8, 42, 37] to great effect.

It is possible to perform contrastive learning for classification in an unsupervised manner, which allows large amounts of unlabeled data to be used [3, 4, 28]. Unsupervised contrastive learning is not possible for deep regression tasks however, because in order for feature representations to be effective for regression, they must reflect label distance relationships in feature space [8, 42, 37]. Contrastive regression loss functions ensure that sample pairs with smaller label distances have features that are

---

*Corresponding author

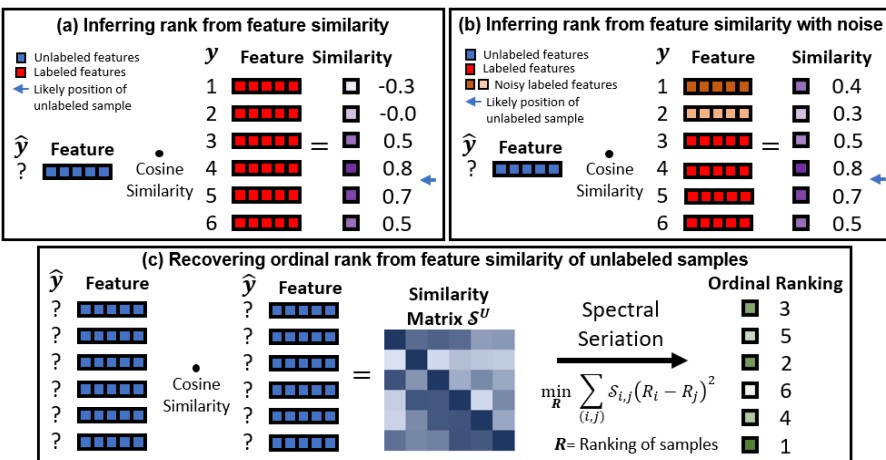

Figure 1: Obtaining rankings from feature similarity. (a) Given labeled samples with well trained features, the ranking of an unlabeled sample can be easily inferred to be between the two with the highest similarity values. (b) The correct ranking can still be inferred with noisy features, as long as the noise levels are within certain thresholds. (c) Given a feature similarity matrix of unlabeled samples, the correct ranking can still be obtained if noise levels are within certain bounds.

more similar, compared to those with greater label distance. Because of this, *existing methods can only be used on fully labeled datasets*, which can be a limiting factor when annotations are costly to obtain. This is especially true for medical imaging analysis, where deep regression is important for estimating real-number medical indicators but often require expert knowledge for manual annotation [8, 20, 40].

In this work, we extend contrastive learning for deep regression *such that unlabeled data can also be used*. We consider a semi-supervised setting where only a small portion of the training dataset is labeled, and the majority of samples are unlabeled. We observe that by enforcing supervised contrastive learning on labeled samples, the feature similarity matrix of unlabeled samples will also learn to reflect label distance [8]. Although the unlabeled feature similarity matrix will be inaccurate and noise corrupted, *it is still possible to infer the relative ordering of samples if errors are within certain bounds*. To give an intuitive example, it is easy to infer the relative ranking of an unlabeled sample based on its pair-wise feature similarity against labeled samples, assuming that features are well-trained (see Fig. 1a). Even in the presence of noisy features, it is still possible to infer the correct ranking if most of the similarity values are reliable (Fig. 1b). Similarly, given a feature similarity matrix for a batch of unlabeled samples, it is still possible to infer their relative ordering even with some degree of noise present (Fig. 1c).

To this end, we propose a novel semi-supervised contrastive regression method by making use of ordinal rankings recovered from the unlabeled feature similarity matrix. We make use of the spectral seriation algorithm proposed by Atkins *et al.* [2] for ranking recovery, which can then be used to construct a distance matrix for supervising contrastive learning on the unlabeled samples. The spectral seriation algorithm is based on an error-minimization approach and allows for some degree of error correction when extracting rankings from the feature similarity matrix. Thus, the ordinal rankings can also be used to supervise predictions on unlabeled samples, which we demonstrate leads to further improvements for semi-supervised regression. We term our method Contrastive Learning with Spectral Seriation (CLSS). Fig. 2 illustrates our overall framework.

We provide theoretical proofs and empirically show that our method can achieve state-of-the-art performance on multiple datasets. Our method is highly beneficial for critical applications such as medical imaging analysis, where annotations can be expensive to obtain. To the best of our knowledge, CLSS is *the first to extend contrastive learning for deep regression to unlabeled data*. To summarize, our main contributions are: (1) we propose CLSS, a novel semi-supervised contrastive learning method that is the first to use unlabeled data for contrastive regression; (2) we demonstrate that spectral seriation can be used to extract robust rankings from the feature similarity matrix of

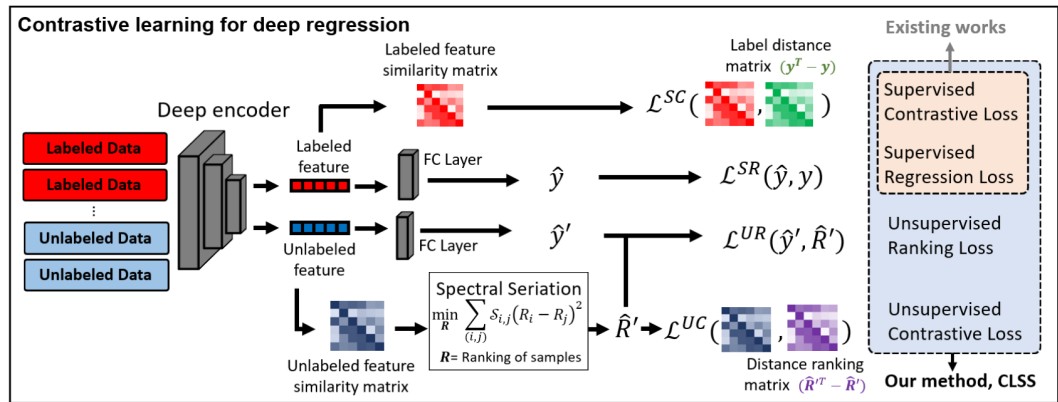

Figure 2: Framework for Contrastive Learning with Spectral Seritation (CLSS). Unlike existing works for contrastive learning that are only able to use labeled data, we make use of spectral seriation to obtain ordinal rankings for unlabeled samples. This can then be used for constrative learning and ranking supervision for unlabeled samples.

unlabeled samples for supervision; (3) we demonstrate that our method can outperform existing state-of-the-art alternatives on different deep regression tasks using multiple datasets.

## 2 Related Work

**Contrastive Learning**

Contrastive learning for classification involves identifying positive and negative sample pairs based on feature similarity, which allows similar samples to be grouped closer together in feature space [3, 19, 15]. These methods can be performed supervised or unsupervised depending on how positive pairs are defined. SimCLR [3] and MoCo [15] defines positive pairs as augmented samples from the same input data, which means labels are not required. Supervised methods such as SupCon [19] define positive pairs as samples belonging to the same class.

Contrastive learning for deep regression requires features to reflect label distance relationships in feature space. Dai *et al.* propose AdaCon, which uses adaptive margins in the SupCon loss function to encourage similarity values to reflect pair-wise label distance [8]. Xue *et al.* [37] find that determining positive and negative pairs based on distance thresholds can be effective for regression tasks. Zhang *et al.* introduce ordinal entropy, which also encourages features to be spread out depending on their pair-wise label distance [42]. Results from these methods show that improving feature learning is a highly effective way to improve deep regression model performance. Because label information is required for these existing contrastive regression loss functions, *they cannot be used with unlabeled data* however. Our method, CLSS, is the first to extend contrastive learning for regression to allow unlabeled data to also be used for training.

**Semi-Supervised Regression**

Semi-supervised learning allows unlabeled data to be used with labeled data for model training. It is an effective way of reducing reliance on manual annotations and is particularly valuable for applications where labeled data is costly to obtain [9, 22, 23, 39, 21]. Semi-supervised classification methods are well studied [29, 41], but semi-supervised deep regression problems receive significantly less attention [7]. Early works such as COREG [44] propose using co-trained KNN neural networks to enforce consistency between two models. Consistency based methods have also been explored in [36, 34]. Iterative approaches for generating pseudo-labels [12], deep kernel learning methods [25, 35, 18], and graph based methods [31] have been proposed for semi-supervised learning in more recent works. A major limitation of these methods is that they are primarily designed for structured tabular data however, and cannot be trained end-to-end with a deep feature extractor for unstructured inputs.

Dai *et al.* proposed UCVME [7], which enforces consistent uncertainty predictions between co-trained models, and can be used for unstructured inputs such as images and videos. Although this

method achieves good performance, it ignores potential improvements that can be made on the feature level, which is critical to the performance of deep regression models [8, 42]. In this work, *we are the first* to address the challenge of performing contrastive learning for deep regression by making use of unlabeled data.

## 3 Method

### 3.1 Overview

Our novel method allows contrastive learning to be performed with *unlabeled data* under a semi-supervised framework. We denote $\mathcal{D} := \{(x_i, y_i)\}_{i=1}^{N}$ as the labeled dataset consisting of $N$ samples, where $x_i$ is the input data and $y_i$ is its corresponding label in $\mathbb{R}$. We denote $\mathcal{D}' := \{x'_{i'}\}_{i'=1}^{N'}$ as the unlabeled dataset consisting of input data only. We denote the feature extractor as $f(\cdot)$ and the feature vector as $z$ for labeled data, such that $z_i = f(x_i)$. Similarly, we denote features for unlabeled data as $z'_i$. Contrastive learning is performed on the L2 normalized feature vector, with some methods also performing feature projection onto a lower dimension [8, 42, 3]. We denote the transformed, normalized feature vector for labeled data as $\tilde{z}$ such that $\tilde{z}_i = \frac{z_i}{||z_i||_2}$. We use $\tilde{z}'$ for unlabeled data. The feature similarity matrix for labeled and unlabeled samples are denoted as $\mathcal{S}$ and $\mathcal{S}'$ respectively.

We denote the regression head as $g(\cdot)$. The overall deep regression model can be expressed as:

$$Y = g(f(X)) + \varepsilon \, ; \;\; \varepsilon \sim N(0, \sigma^2) \, . \tag{1}$$

The model can be trained through supervised regression by minimizing mean squared error (MSE) loss, which we denote as $\mathcal{L}^{SR}$:

$$\mathcal{L}^{SR} = \frac{1}{N} \sum_{i=1}^{N} (y_i - \hat{y}_i)^2 \, . \tag{2}$$

Supervised contrastive learning can be performed on the normalized feature vectors for labeled samples $\tilde{z}$ using different contrastive regression losses, which we denote as $\mathcal{L}^{SC}$. Spectral seriation can then be used to extract the ordinal rankings $R'$ from $\mathcal{S}'$ for supervising contrastive learning and predictions on labeled samples. We describe our method in detail below.

### 3.2 Using ordinal ranking from spectral seriation for supervision

#### 3.2.1 Spectral seriation for retrieving ordinal ranking from similarity matrix

Contrastive learning for deep regression aims to ensure that feature representations reflect label distance relationships in feature space [8, 42, 37]. If $y_i$ and $y_j$ are closer together than $y_i$ and $y_k$, than the distance between features $z_i$ and $z_j$ should be closer together than $z_i$ and $z_k$. Thus:

$$||\tilde{z}_i - \tilde{z}_j||_2 < ||\tilde{z}_i - \tilde{z}_k||_2 \;\; \text{for} \;\; |y_i - y_j| < |y_i - y_k| \, . \tag{3}$$

Alternatively, because we use L2 normalized feature vectors,

$$\gamma(\tilde{z}_i, \tilde{z}_j) > \gamma(\tilde{z}_i, \tilde{z}_k) \;\; \text{for} \;\; |y_i - y_j| < |y_i - y_k| \, , \tag{4}$$

where $\gamma$ is the cosine similarity function, and $\gamma(\tilde{z}_i, \tilde{z}_j)$ and $\gamma(\tilde{z}_i, \tilde{z}_k)$ correspond to entries in similarity matrix $\mathcal{S}$. By performing supervised contrastive learning on labeled samples, we can also expect entries in $\mathcal{S}'$ to also approximate Eq. 4 with noise, as they have been trained to reflect distance relationships in feature space.

To retrieve the ordinal ranking $R'$ of samples in an unlabeled batch, we can make use of the spectral seriation algorithm originally proposed by Atkins *et al.* [2]. The algorithm is designed to recover an ordinal ranking given a correlation matrix, where higher correlation values indicate closer proximity in ranking. The algorithm is especially useful for when it is easy to determine the similarity between sample pairs but difficult to obtain a direct ordering. Cosine similarity is equivalent to correlation after L2 normalization. Therefore spectral seriation can also be used to recover $R'$ from $\mathcal{S}'$.

The seriation problem can be formulated as a loss minimizing function:

$$\underset{R'}{\arg\min} \sum_{i,j} \mathcal{S}'_{i,j} (R'_i - R'_j)^2 \, , \tag{5}$$

where $\mathcal{S}'_{i,j} = \gamma(\tilde{z}_i, \tilde{z}_j)$. Since sample pairs closer together in ranking have higher correlation values, minimizing this loss encourages these samples to have $R_i$ and $R_j$ that are closer together. As per spectral seriation, the solution to $R'$ can be derived from the Fiedler vector, as stated in Theorem 1.

**Theorem 1** *Given similarity matrix $\mathcal{S}'$ such that $\mathcal{S}'_{i,j} > \mathcal{S}'_{i,k}$ for $|y_i - y_j| < |y_i - y_k|$, the ordinal ranking that best satisfies observed $\mathcal{S}'$ is the ranking of the values in the Fiedler vector of $\mathbf{L}$, where $\mathbf{L}$ is the Laplacian of $\mathcal{S}'_{i,j}$.*

Intuitively, the proof of this theorem can be obtained by approximating discrete rankings $R'$ with real-number values $r'$ and expressing Eq. 5 in the form of:

$$\min_{r'^T e = 0, r'^T r' = 1} r'^T \mathbf{L} r' \ . \tag{6}$$

$r'$ can be solved by computing the Fiedler vector, which is the eigenvector corresponding with the smallest non-zero eigenvalue. The relative rankings of $r'$ then give us $R'$. We refer interested readers to [2] for detailed derivations.

### 3.2.2 Unlabeled contrastive learning with seriation rankings

Spectral seriation gives us rank $R'$ of unlabeled samples within a batch. From this, we can obtain rankings on the distances between sample pairs for some anchor sample $i$. The distance ranking can be used to supervise contrastive learning on unlabeled samples by ensuring that the feature similarity values are consistent with the rankings. We define the unlabeled contrastive learning loss with respect to some subset $\mathcal{B}$ as:

$$\mathcal{L}^{UC} = \sum_{i=1}^{|\mathcal{B}|} \ell \left( \mathbf{rk}(\mathcal{S}'_{[i,:]}), \ \mathbf{rk}(-|R' - R'_{[i]}|); \lambda \right), \tag{7}$$

where $[i,:]$ denotes the $i$th row in the matrix, $[i]$ denotes the $i$th value of a vector, $\mathbf{rk}$ denotes the ranking operator, and $\ell$ is the ranking similarity function. This loss function ensures that the ranking of the feature similarity values with anchor sample $i$ are consistent with that of the distances between derived rankings obtained from seriation. For the differentiable ranking similarity loss function $\ell$, we directly use the differential combinatorial solver proposed in [27], which takes an additional parameter $\lambda$. On a high level, the function makes use of interpolation methods to allow combinatorial inputs to be differentiable, although detailed explanations can be found in the original work.

### 3.2.3 Ranking supervision of unlabeled predictions with seriation rankings

Given $R'$ for unlabeled samples, we can also perform supervision on the prediction output $\hat{y}$. The spectral seriation algorithm is inherently robust to error, which we show in section 3.3, which means pair-wise distance rankings from seriation are likely to be more accurate than the predicted output. We define the unlabeled prediction ranking loss with respect to some subset $\mathcal{B}$ as:

$$\mathcal{L}^{UR} = \sum_{i=1}^{|\mathcal{B}|} \ell \left( \mathbf{rk}(-|\hat{y}' - \hat{y}'_{[i]}|), \ \mathbf{rk}(-|R' - R'_{[i]}|); \lambda \right), \tag{8}$$

where $[i]$ denotes the $i$th value of a vector. This serves as an additional supervision for unlabeled data.

### 3.2.4 Overall framework

The total loss function $\mathcal{L}$ of our method is:

$$\mathcal{L} = \mathcal{L}^{SR} + w_{SC}\mathcal{L}^{SC} + w_{UC}\mathcal{L}^{UC} + w_{UR}\mathcal{L}^{UR} \ , \tag{9}$$

where $\mathcal{L}^{SR}$, $\mathcal{L}^{SC}$, $\mathcal{L}^{UC}$, and $\mathcal{L}^{UR}$ represent the loss values of supervised regression, supervised contrastive loss, unsupervised contrastive loss, and unsupervised ranking loss. $w_{SC}$, $w_{UC}$, and $w_{UR}$ are the corresponding loss weights. $\mathcal{L}^{UC}$ and $\mathcal{L}^{UR}$ are calculated using unlabeled data with the ordinal rankings recovered through spectral seriation. Fig. 2 shows the overall framework.

### 3.3 Robustness analysis of spectral seriation

Because spectral seriation is based on a loss minimization approach, the algorithm itself is robust to noisy inputs. In this section, we provide theoretical proofs to formally show that spectral seriation is robust to two different types of noise: noisy similarity matrices, and noisy feature representations. We derive approximate bounds to analytically demonstrate that spectral seriation can be a reliable method for recovering ordinal rankings from noisy similarity values of unlabeled samples. Additional derivations are also included in the supplementary materials for more technical readers.

#### 3.3.1 Robustness to noise in the similarity matrix

We make use of matrix perturbation theory to show that spectral seriation ranking is robust to noisy values in similarity matrix $\mathcal{S}'$. We give an upper bound for the noisy entries of the similarity matrix, and show that when the errors of the similarity matrix are bounded, our spectral ranking algorithm recovers the true ranking, thereby demonstrating the error correcting nature of spectral seriation.

The main result is presented in Theorem 2. We first present two related lemmas to assist with the proof, where Lemma 1 provides the perturbation bounds for eigenvalues of symmetric matrices, and Lemma 2 provides the upper bound of the Fiedler value.

**Lemma 1** *Let* $\mathbf{A}, \mathbf{B} \in \mathbb{C}^{n \times n}$ *be Hermitian matrices,* $\lambda(\mathbf{A}) = \{\lambda_i\}, \lambda(\mathbf{B}) = \{\mu_i\}, \lambda_1 \leq \lambda_2 \leq \cdots \leq \lambda_n, \mu_1 \leq \mu_2 \leq \cdots \leq \mu_n$, *then:*

$$|\mu_i - \lambda_i| \leq \|\mathbf{B} - \mathbf{A}\|_2.$$

**Lemma 2** $\lambda$ *is the Fiedler value of the Laplacian matrix* $\mathbf{L}$ *of the similarity matrix* $\mathcal{S}'$, *then:*

$$\lambda \leq \tfrac{n}{n-1} \min_{1 \leq i \leq n} \{\mathbf{L}_{ii}\}.$$

**Theorem 2** *For a similarity matrix* $\mathcal{S}'$, *suppose the error matrix of it is* $\Delta\mathcal{S}'$. *When* $2\|\Delta\mathcal{S}'\|_F \leq 1 - \frac{\min_{1 \leq i \leq n}\left\{\sum_{t \neq i} |S'_{it}|\right\}}{n-1}$, *the seriation obtained by the spectral ranking algorithm using* $\mathcal{S}'$ *is the same as that obtained by the spectral ranking algorithm using* $\mathcal{S}' + \Delta\mathcal{S}'$.

The proof of Theorem 2 relies mainly on the definition of the Fielder vector, Lemma 1 and Lemma 2. First order approximation is used to approximate changes in the Fielder vector after adding noise to the similarity matrix, thus allowing us to obtain upper bounds for the noise level. Detailed derivations are included in the supplementary materials.

#### 3.3.2 Robustness to noise in feature representations

We next consider the case where the feature representation for some given sample $\tilde{z}'_i$ is noisy. The direct consequence of this is that all entries in rows $i$ and column $j$ of $\mathcal{S}'$ will be noisy. We can also use a similar approach to show that the spectral seriation algorithm can recover the correct ranking as long as the error values are within an upper bound.

**Theorem 3** *For a similarity matrix* $\mathcal{S}'$, *suppose rows* $i$ *and column* $i$ *are corrupted due to inadequate feature representations being learnt for sample* $i$. *When* $\|\Delta\boldsymbol{S}'_{[i,:]}\|_2 - \|\Delta\boldsymbol{S}'_{[i,:]}\|_1 + \max_{1 \leq j \leq n} |\Delta\mathcal{S}'_{ij}| \leq 1 - \frac{\min_{1 \leq i \leq n}\left\{\sum_{t \neq i} |S'_{it}|\right\}}{n-1}$, *the seriation obtained by the spectral ranking algorithm using* $\mathcal{S}'$ *is the same as that obtained by the spectral ranking algorithm using* $\mathcal{S}' + \Delta\mathcal{S}'$.

The proof for Theorem 3 can be obtained following a similar approach to Theorem 2. We include detailed derivations in the supplementary materials.

## 4 Experiments

We evaluate our proposed method using three different types of datasets to demonstrate its effectiveness as a general approach for semi-supervised deep regression. We use a synthetic non-linear dataset for operator learning, a medical imaging dataset for brain age estimation from MRI scans, and a natural image dataset for age estimation from photographs.

## 4.1 Synthetic dataset for non-linear operator learning

We use the non-linear synthetic dataset generated by Lu *et al.* in problem 6 of [24] and train a neural network to estimate the operator function. The target is the stochastic partial differential equation:

$$-\text{div}(e^{b(x;w)}\nabla u(x;w)) = f(x) , \tag{10}$$

where $w$ is stochastic, $x \in (0,1)$, and $e^{b(x;w)}$ is a diffusion coefficient where $b(x;w)$ follows a random Gaussian process. The Dirichlet boundary conditions are $u(0) = u(1) = 0$, and $f(x) = 10$. The input data $\{x\}_{i=1}^N$ are outputs generated by $(x;w)$ and the target label $\{y\}_{i=1}^N$ is the solution of $u(x;w)$. More details of the data generation process can be found in problem 6 of [24].

We use the same architecture and training scheme following [42], which consists of a two-layer fully connected neural network with 100 hidden units. We perform 10 separate training runs on 1,000 samples each and test our model on a set of 100,000 samples. Mean and standard deviation of the 10 runs are reported. We use ordinal entropy [42] as the contrastive loss function $\mathcal{L}^{SC}$, which is imposed on the feature layer after L2 normalization. Training is performed with a learning rate of $1 \times 10^{-3}$. We use the entire dataset as an input batch and train for 100,000 epochs. Smaller batches of 10 samples are used to calculate $\mathcal{L}^{SC}$, $\mathcal{L}^{UC}$, and $\mathcal{L}^{UR}$ to reduce computation time due to the quadratic scaling of the feature similarity matrix. We set $w_{SC}, w_{UC}$, and $w_{UR}$ to $1 \times 10^{-3}$ and $\lambda$ to 2. Implementation is done in PyTorch and training is performed on a single V100 Nvidia GPU.

### 4.1.1 Comparison with state-of-the-art alternatives

We compare with state-of-the-art semi-supervised deep regression methods to demonstrate the effectiveness of our method, CLSS. We adapt conventional mean-teacher [30] *(Mean-teacher)* and cross psuedo-label [5] supervision (*CPS*) semi-supervised learning methods for deep regression using a single output value as the regression prediction. We also compare with the state-of-the-art method, UCVME, proposed in [7]. For reference, we show results using a supervised naïve regression method using only labeled data. We show in Table 1 results for different settings, where 1/5, 1/4, 1/3, and 1/2 of available labels are used, and the remaining samples are treated as unlabeled data. Additional implementation details are provided in the supplementary materials section.

Table 1: Comparison with state-of-the-art methods on synthetic non-linear dataset.

MAE↓

| Type | Method | 1/5 labels | 1/4 labels | 1/3 labels | 1/2 labels |
|------|--------|-----------|-----------|-----------|-----------|
| *Supervised* | Regression | $0.098 \pm 0.095$ | $0.056 \pm 0.016$ | $0.041 \pm 0.015$ | $0.032 \pm 0.009$ |
| *Semi-supervised* | Mean-teacher [30] | $0.080 \pm 0.089$ | $0.047 \pm 0.021$ | $0.043 \pm 0.019$ | $0.029 \pm 0.011$ |
| | CPS [5] | $0.057 \pm 0.012$ | $0.045 \pm 0.016$ | $0.041 \pm 0.015$ | $0.028 \pm 0.007$ |
| | UCVME [7] | $0.040 \pm 0.008$ | $0.033 \pm 0.008$ | $0.027 \pm 0.007$ | $0.028 \pm 0.021$ |
| | CLSS (Ours) | $\mathbf{0.033 \pm 0.008}$ | $\mathbf{0.027 \pm 0.009}$ | $\mathbf{0.020 \pm 0.007}$ | $\mathbf{0.016 \pm 0.007}$ |

$\mathbf{R^2}$↑

| Type | Method | 1/5 labels | 1/4 labels | 1/3 labels | 1/2 labels |
|------|--------|-----------|-----------|-----------|-----------|
| *Supervised* | Regression | $66.9\% \pm 39.4$ | $83.8\% \pm 7.7$ | $88.5\% \pm 8.0$ | $90.9\% \pm 5.1$ |
| *Semi-supervised* | Mean-teacher [30] | $69.4\% \pm 40.1$ | $86.9\% \pm 8.4$ | $90.7\% \pm 7.6$ | $92.5\% \pm 7.2$ |
| | CPS [5] | $84.5\% \pm 8.8$ | $88.8\% \pm 8.5$ | $88.5\% \pm 8.0$ | $93.3\% \pm 5.3$ |
| | UCVME [7] | $92.2\% \pm 3.6$ | $94.2\% \pm 2.8$ | $95.0\% \pm 3.0$ | $95.6\% \pm 4.3$ |
| | CLSS (Ours) | $\mathbf{96.4\% \pm 1.7}$ | $\mathbf{97.3\% \pm 2.4}$ | $\mathbf{98.4\% \pm 1.3}$ | $\mathbf{99.3\% \pm 0.5}$ |

We can see that semi-supervised methods generally outperform naïve supervised regression. Our proposed method, CLSS, convincingly outperforms alternative methods however and consistently improves $\mathbf{R}^2$ by 3-4% over the next best alternative across all settings. Computational and memory costs for each method are also provided in Section S3.1.5 of the supplementary materials for reference.

### 4.1.2 Ablation Study

To analyse the effect of different components in our methodology, namely the use of $\mathcal{L}^{SC}$, $\mathcal{L}^{UC}$, and $\mathcal{L}^{UR}$, we perform training with the loss functions added separately to study their impact. We plot the results in Fig. 3 for easier visualization. We can see that each individual component leads to

significant contributions in improved performance across all settings. This provides further empirical support of the effectiveness of CLSS.

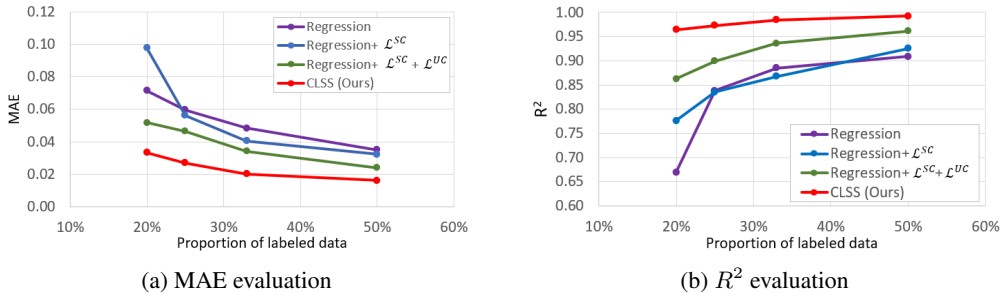

(a) MAE evaluation            (b) $R^2$ evaluation

Figure 3: Ablation results from adding $\mathcal{L}^{SC}$, $\mathcal{L}^{UC}$ and $\mathcal{L}^{UR}$ separately. Using $\mathcal{L}^{UC}$ and $\mathcal{L}^{UR}$ to supervise unlabeled samples consistently leads to better predictions.

### 4.1.3 Quality of ordinal rankings from spectral seriation

CLSS uses the ordinal rankings recovered from $\mathcal{S}'$ to supervise predictions on unlabeled samples. This is based on the observation that the spectral seriation algorithm is robust to errors and more likely to provide accurate supervision. To validate this, we compare with results using rankings derived from predictions $\hat{y}$ instead of from spectral seriation. We formulate a new loss $\mathcal{L}^{UCP}$:

$$\mathcal{L}^{UCP} = \sum_{i=1}^{|\mathcal{B}|} \ell\left(\mathbf{rk}(\mathcal{S}'_{[i,:]}), \, \mathbf{rk}(-|\hat{y}' - \hat{y}'_{[i]}|)\right), \tag{11}$$

and train the model using loss function $\mathcal{L} = \mathcal{L}^{SR} + w_{SC}\mathcal{L}^{SC} + w_{UCP}\mathcal{L}^{UCP}$. We do not include $\mathcal{L}^{UR}$ since predictions $\hat{y}$ are already used to derive the ranking. Results are shown in Table 2.

Table 2: Results using different rankings for supervision

| Method | MAE $\downarrow$ / $\mathbf{R}^2\uparrow$ | | | |
|---|---|---|---|---|
| | 1/5 labels | 1/4 labels | 1/3 labels | 1/2 labels |
| Regression+$\mathcal{L}^{SC}$+$\mathcal{L}^{UCP}$ | 0.039 / 92.4% | 0.030 / 95.8% | 0.024 / 96.8% | 0.016 / 98.5% |
| CLSS (Ours) | **0.033 / 96.4%** | **0.027 / 97.3%** | **0.020 / 98.4%** | **0.016 / 99.3%** |

We can see that CLSS consistently performs better than using rankings derived from predictions $\hat{y}'$. This validates the observation that rankings obtained through spectral seriation are more robust, leading to better supervision on unlabeled samples.

### 4.2 Validation on Brain Age estimation from MRI Scans

Semi-supervised deep regression problems are particularly valuable for medical applications, as real-number medical indicators are common for disease tracking [8, 26, 17], and labeled data are usually costly to obtain [20, 21, 9]. We validate CLSS on the IXI brain MRI dataset for brain age estimation [13]. Brain age estimation involves training a model to learn relevant phenotypes from MRI scans associated with brain health and has important applications in detection of diseases such as Alzeimer's [6, 14]. Typically, a model is trained using brain MRIs from healthy patients to predict *chronological* age, and then used to make predictions for unhealthy patients to estimate their *biological* brain age for disease screening [1]. This process assumes that patients used for training have similar chronological and biological ages however, which requires data from healthy individuals to be used. By making use of unlabeled data, we can make training more robust since chronological age is not strictly enforced as the ground truth for these samples. This also reduces the reliance on healthy patients for training.

The IXI dataset consists of 588 MRI brain scans with corresponding ages between 20 and 86 [13]. Out of these, we use 88 samples as the test set and 80 samples as the validation set. A 3D ResNet-18 [32] is used as our model. We use a learning rate of $1 \times 10^{-3}$ with 0.1 decay every 10 epochs and

train for a total of 30 epochs. We use a batch size of 16 for labeled samples and a batch size of 8 for unlabeled samples. We set $w_{SC} = 1$, $w_{UC} = 0.05$, $w_{UR} = 0.01$, and $\lambda = 2$, which were chosen based on the validation set. To account for unstable training, all experiments were run separately 10 times using different random seeds. Mean and standard deviation of the 10 runs are reported. We use PyTorch for implementation and train on a single V100 Nvidia GPU.

### 4.2.1 Comparison with state-of-the-art alternatives and ablation studies

We compare results from using CLSS with alternative state-of-the-art semi-supervised methods and show results in Table 3. We use settings where 1/5, 1/4, 1/3, and 1/2 of the dataset is treated as labeled data and remaining samples are treated as unlabeled data.

Table 3: Comparison with state-of-the-art methods on IXI brain age dataset.

| Type | Method | MAE↓ | | | |
| | | 1/5 labels | 1/4 labels | 1/3 labels | 1/2 labels |
|---|---|---|---|---|---|
| *Supervised* | Regression | $9.95 \pm 1.41$ | $11.93 \pm 1.40$ | $11.76 \pm 1.75$ | $10.93 \pm 1.60$ |
| *Semi-supervised* | Mean-teacher [30] | $11.23 \pm 2.31$ | $10.27 \pm 1.57$ | $10.52 \pm 3.12$ | $12.01 \pm 2.03$ |
| | CPS [5] | $10.23 \pm 1.41$ | $10.27 \pm 1.19$ | $\mathbf{9.64 \pm 1.27}$ | $9.69 \pm 1.01$ |
| | UCVME [7] | $9.83 \pm 1.32$ | $10.86 \pm 1.67$ | $9.65 \pm 1.31$ | $10.06 \pm 1.19$ |
| | CLSS (Ours) | $\mathbf{9.58 \pm 1.48}$ | $\mathbf{9.68 \pm 1.22}$ | $9.72 \pm 1.29$ | $\mathbf{9.37 \pm 1.17}$ |

We can see from the results that our method, CLSS, performs the best under all semi-supervised settings except 1/3 labels. CLSS therefore can be used as an effective way of reducing reliance on labeled data for medical imaging analysis applications. Additional ablation experiments are included in the supplementary materials and demonstrate the importance of each component in CLSS.

## 4.3 Validation on Age-Estimation from photographs

We also validate our method on a natural image dataset to provide further empirical support. We use the AgeDB-DIR dataset [38] for performing age-estimation from photographs, a common benchmark task for deep regression. Although photographs of people can easily be obtained online, accurate age labels are not always available due to privacy issues. This challenge can be addressed through semi-supervised deep regression methods to reduce reliance on labeled data. AgeDB-DIR consists of 16,488 images of people with ages ranging between 1 and 101. The dataset has fewer tail samples to reflect real-world label imbalance settings. We use the same data splits provided by the dataset for training, validation, and testing.

We use a ResNet50 network [16] pretrained on ImageNet [11] as our deep regression model. We use a learning rate of $5 \times 10^{-4}$ with decay of 0.1 every 10 epochs and train for 30 epochs. We use a batch size of 32 for labeled samples and a batch size of 8 for unlabeled samples. We set $w_{SC} = 1$, $w_{UC} = 0.05$, $w_{UR} = 0.01$, and $\lambda = 2$, which were chosen based on the validation set. To account for unstable training, all experiments were run separately 10 times using different random seeds. Mean and standard deviation of the 10 runs are reported. We use PyTorch for implementation and train on a single V100 Nvidia GPU.

### 4.3.1 Comparison with state-of-the-art alternatives and ablation studies

We compare results from using CLSS with alternative state-of-the-art semi-supervised methods and show results in Table 4. We use settings where 1/30, 1/25, 1/20, and 1/15 of the dataset is treated as labeled data and remaining samples are treated as unlabeled data.

We can see from the results that CLSS outperforms alternative methods for all settings. Overall, the experiments demonstrate that CLSS can also be applied effectively to natural image datasets. We also include ablation experiments in the supplementary materials to highlight the effect of different components.

Table 4: Comparison with state-of-the-art methods on AgeDB-DIR dataset.

| Type | Method | 1/30 labels | 1/25 labels | 1/20 labels | 1/15 labels |
|------|--------|-------------|-------------|-------------|-------------|
| *Supervised* | Regression | $10.14 \pm 0.25$ | $9.99 \pm 0.11$ | $9.10 \pm 0.15$ | $8.58 \pm 0.10$ |
| *Semi-supervised* | Mean-teacher [30] | $10.05 \pm 0.29$ | $9.99 \pm 0.13$ | $9.05 \pm 0.12$ | $8.62 \pm 0.09$ |
| | CPS [5] | $9.99 \pm 0.12$ | $9.83 \pm 0.10$ | $8.99 \pm 0.14$ | $8.47 \pm 0.08$ |
| | Ours | $\mathbf{9.95 \pm 0.18}$ | $\mathbf{9.59 \pm 0.12}$ | $\mathbf{8.88 \pm 0.09}$ | $\mathbf{8.45 \pm 0.11}$ |

MAE↓

# 5 Conclusion

In this work, we propose a novel approach, CLSS, that allows unlabeled data to be used for contrastive learning on deep regression tasks. We make use of the observation that the feature similarity matrix of unlabeled samples also reflect label distance between samples, and that a robust ordinal ranking can be extracted from the matrix using spectral seriation. We derive theoretical bounds for error values in the similarity matrix for which the derived ordinal ranking remains correct, thereby demonstrating the robustness of our method. We validate our method empirically on a synthetic dataset and two real-world datasets and show that our method can outperform alternative state-of-the-art semi-supervised deep regression methods. Overall, CLSS is a useful technique for improving the performance of semi-supervised deep regression models.

# 6 Acknowledgements

This research is supported by grants from the National Natural Science Foundation of China/HKSAR Research Grants Council Joint Research Scheme under Grant N_HKUST627/20, by the Project of Hetao Shenzhen-Hong Kong Science and Technology Innovation Cooperation Zone (HZQB-KCZYB-2020083), by the Hong Kong Innovation and Technology Commission (Project no. ITS/030/21 & Project no. PRP/041/22FX), and by Foshan HKUST Projects under FSUST21-HKUST10E and FSUST21-HKUST11E.

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
