# Supplementary Materials:
# Semi-Supervised Contrastive Learning for Deep Regression with Ordinal Rankings from Spectral Seriation

**Weihang Dai[1], Yao Du[1], Hanru Bai[3], Kwang-Ting Cheng[1], Xiaomeng Li[1,2]***
[1]The Hong Kong University of Science and Technology
[2]HKUST Shenzhen-Hong Kong Collaborative Innovation Research Institute, Futian, Shenzhen
[3]Fudan University
eexmli@ust.hk

### S2.3.1  Robustness to noise in the similarity matrix

**Proof via First-Order Approximation**

We derive approximate bounds for error tolerance using a first-order approximation approach to theoretically illustrate the robustness of spectral seriation. The main result is presented in Theorem 2. We first present two related lemmas to assist with the proof, where Lemma 1 provides the perturbation bounds for eigenvalues of symmetric matrices, and Lemma 2 provides the upper bound of the Fiedler value.

**Lemma 1** *Let* $\mathbf{A}, \mathbf{B} \in \mathbb{C}^{n \times n}$ *be Hermitian matrices,* $\lambda(\mathbf{A}) = \{\lambda_i\}, \lambda(\mathbf{B}) = \{\mu_i\}, \lambda_1 \leq \lambda_2 \leq \cdots \leq \lambda_n, \mu_1 \leq \mu_2 \leq \cdots \leq \mu_n$, *then:*

$$|\mu_i - \lambda_i| \leq \|\mathbf{B} - \mathbf{A}\|_2.$$

**Lemma 2** $\lambda$ *is the Fiedler value of the Laplacian matrix* $\mathbf{L}$ *of the similarity matrix* $\mathcal{S}'$, *then:*

$$\lambda \leq \frac{n}{n-1} \min_{1 \leq i \leq n} \{\mathbf{L}_{ii}\}.$$

**Theorem 2** *For a similarity matrix* $\mathcal{S}'$, *suppose the error matrix of it is* $\Delta \mathcal{S}'$. *When* $2\|\Delta \mathcal{S}'\|_F \leq 1 - \frac{\min_{1 \leq i \leq n} \{\sum_{t \neq i} |\mathcal{S}'_{it}|\}}{n-1}$, *the seriation obtained by the spectral ranking algorithm using* $\mathcal{S}'$ *is the same as that obtained by the spectral ranking algorithm using* $\mathcal{S}' + \Delta \mathcal{S}'$.

*Proof.* To prove the above statement, it is sufficient to demonstrate that when $f_i \geq f_j$, $f_i + \Delta f_i \geq f_j + \Delta f_j$, $\sum_{t \neq i} \Delta \mathcal{S}'_{it} \leq \sum_{t \neq j} \Delta \mathcal{S}'_{jt}$, that is,

$$f_i - f_j \geq \Delta f_j - \Delta f_i, \forall 1 \leq i, j \leq n, \tag{12}$$

where $f_i$ is the $i$-th element of the Fiedler vector $\mathbf{f}$ of $\mathcal{S}'$, $\Delta f_i$ is the change in $f_i$ after adding noise.

The following proof is only considering the situation where $f_i, f_j \geq 0, \Delta f_i \leq 0, \Delta f_j \geq 0$. When the given upper bound of the error satisfies the above condition, the other conditions are also satisfied.

According to the definition of the Fiedler vector, we have

$$(\mathbf{L} + \Delta \mathbf{L})(f + \Delta f) = (\lambda + \Delta \lambda)(f + \Delta f). \tag{13}$$

---

*Corresponding author

37th Conference on Neural Information Processing Systems (NeurIPS 2023).

We make a simplified approximation by expanding the above equation and neglect terms that approach zero in most cases, then only considering the $i$-th component, we have

$$\left(\sum_{t\neq i}\mathcal{S}'_{it}-\lambda+1\right)\Delta f_i\approx\left(\Delta\lambda-\sum_{t\neq i}\Delta\mathcal{S}'_{it}\right)f_i. \tag{14}$$

According to Lemma 2,

$$\Delta f_i\geq\frac{n-1}{n-1-\min_{1\leq i\leq n}\left\{\sum_{t\neq i}|S'_{it}|\right\}}\left(\Delta\lambda-\sum_{t\neq i}\Delta\mathcal{S}'_{it}\right)f_i, \tag{15}$$

$$\Delta f_j\leq\frac{n-1}{n-1-\min_{1\leq i\leq n}\left\{\sum_{t\neq i}|S'_{it}|\right\}}\left(\Delta\lambda-\sum_{t\neq j}\Delta\mathcal{S}'_{jt}\right)f_j. \tag{16}$$

Then,

$$\Delta f_j-\Delta f_i\leq\frac{n-1}{n-1-\min_{1\leq i\leq n}\left\{\sum_{t\neq i}|S'_{it}|\right\}}\left[-\Delta\lambda+(n-1)\max_{1\leq i,j\leq n}\left|\Delta\mathcal{S}'_{ij}\right|\right](f_i-f_j) \tag{17}$$

$$\leq f_i-f_j \tag{18}$$

Combined with Lemma 1, and $||\cdot||_2\leq||\cdot||_{\mathbf{F}}$, we have

$$2||\Delta\mathcal{S}'||_F\leq 1-\frac{\min_{1\leq i\leq n}\left\{\sum_{t\neq i}|S'_{it}|\right\}}{n-1}. \tag{19}$$

$\square$

It is noted that during the process of proof, we assume that the diagonal elements of $\mathcal{S}'$ are not perturbed. This is reasonable since the diagonal elements correspond to cosine similarity between the same features, which will always be 1. This proof demonstrates that the Fiedler vector is tolerant to error values in $\mathcal{S}'$.

**Proof Via Eigenvalue gaps**

We can also derive tighter bounds for error tolerance using a more rigorous approach via eigenvalue gap analysis. We outline the proof below for interested readers.

The main result is presented in Theorem 2. We first present Stewart's theorem in Lemma 1 to assist with the proof, where it can provide corresponding eigenvalue conditions for the stability of the subspace spanned by the eigenvectors.

**Lemma 1 (Stewart's theorem).** *Let $S, E\in\mathbb{R}^{n\times n}$ be symmetric matrices and consider $V_1\in\mathbb{R}^{d\times n}$, $V_2\in\mathbb{R}^{(n-d)\times n}$, where range($V_1$) is an invariant subspace for $S$. Let $V=[V_1,V_2]$ be an orthogonal matrix, and let:*

$$V^TSV=\left[\begin{array}{cc}Q_1 & 0\\0 & Q_2\end{array}\right],V^TEV=\left[\begin{array}{cc}E_{11} & E_{12}\\E_{21} & E_{22}\end{array}\right].$$

*If $\delta=\lambda_{\min}-\mu_{\max}-\|E_{11}\|_2-\|E_{22}\|_2>0$, and $\|E_{21}\|_2\leq\frac{\delta}{2}$, where $\lambda$ and $\mu$ are the eigenvalues of $Q_1$ and $Q_2$, respectively, then there exists a matrix $P\in\mathbb{R}^{(n-d)\times n}$ with $\|P\|_2\leq\frac{2}{\delta}\|E_{21}\|_2$ such that the columns of $V_1'=(V_1+V_2P)\left(I+P^TP\right)^{\frac{1}{2}}$ form an invariant subspace for $S+E$.*

**Theorem 2** *For a similarity matrix $\mathcal{S}'\in\mathbb{R}^{n\times n}$, suppose the error matrix of it is $E\in\mathbb{R}^{n\times n}$. When*

$$||E||_1\leq\tfrac{\lambda_3-\lambda_2}{8\sqrt{n}},$$

where $\lambda_2, \lambda_3$ are the second smallest and the third smallest eigenvalue of Laplacian matrix of $\mathcal{S}'$, the Fiedler vector of $\mathcal{S}' \in \mathbb{R}^{n \times n}$ is stable. So the seriation obtained by the spectral ranking algorithm is robust to noise in $\mathcal{S}'$.

*Proof.* Considering that when eigenvalues cluster together, the corresponding eigenvectors are ill conditioned and the inappropriateness of using eigenvectors can also be observed when dealing with a matrix having two eigenvectors with equal eigenvalues, in this section, we begin by studying the stability of the subspace spanned by Fiedler vector and unit vector, thereby ensuring the stability of Fiedler vector naturally.

Let the eigenvalues of Laplacian matrix $L^{(\mathcal{S}')}$ and $L^{(E)}$ be $\lambda_n \geq \lambda_{n-1} \geq \cdots \geq \lambda_1$ and $\epsilon_n \geq \epsilon_{n-1} \geq \cdots \geq \epsilon_1$ respectively. Correspondingly, the eigenvalues of $-L^{(\mathcal{S}')}$ and $-L^{(E)}$ are $-\lambda_i$ and $-\epsilon_i$, $i = 1, \ldots, n$, and they share the same eigenvectors with $L^{(\mathcal{S}')}$ and $L^{(E)}$ respectively. Thus, Fiedler vector and unit vector are also eigenvectors of $-L^{(\mathcal{S}')}$.

We have

$$-\lambda_2 - (-\lambda_3) - \left\|L_{11}^{(E)}\right\|_2 - \left\|L_{22}^{(E)}\right\|_2 \geq -\lambda_2 - (-\lambda_3) - 2\|L^{(E)}\|_2. \tag{20}$$

When $-\lambda_2 - (-\lambda_3) > 4\|L^{(E)}\|_2$, that is, $-\lambda_2 - (-\lambda_3) - 2\|L^{(E)}\|_2 > 2\|L^{(E)}\|_2$. Then equation (12) becomes

$$-\lambda_2 - (-\lambda_3) - \left\|L_{11}^{(E)}\right\|_2 - \left\|L_{22}^{(E)}\right\|_2 > 2\|L^{(E)}\|_2 \geq 2\left\|L_{12}^{(E)}\right\|_2. \tag{21}$$

Based on equation (13), we can derive $\delta \geq 2\left\|L_{12}^{(E)}\right\|_2$. Next, it is obviously that when $\|L^{(E)}\|_2 > 0$, $\delta > 0$.

Furthermore, we can derive that

$$\|L^{(E)}\|_2 \leq \sqrt{n}\|L^{(E)}\|_1 \leq 2\sqrt{n}\|E\|_1. \tag{22}$$

To sum up, when $8\sqrt{n}\|E\|_1 \leq \lambda_3 - \lambda_2$, the prerequisites of Stewart's theorem are met, so the Fiedler vector is stable under perturbation $E$. So the seriation obtained by the spectral ranking algorithm is robust to noise in $\mathcal{S}'$.

$\square$

### S2.3.2  Robustness to noise in feature representations

**Proof Via First-Order Approximation**

**Theorem 3** *For a similarity matrix $\mathcal{S}'$, suppose rows $i$ and column $i$ are corrupted due to inadequate feature representations being learnt for sample $i$. When $\|\Delta \boldsymbol{S}'_{[i,:]}\|_2 - \|\Delta \boldsymbol{S}'_{[i,:]}\|_1 + \max_{1 \leq j \leq n} \left|\Delta \mathcal{S}'_{ij}\right| \leq 1 - \frac{\min_{1 \leq i \leq n}\left\{\sum_{t \neq i} |S'_{it}|\right\}}{n-1}$, the seriation obtained by the spectral ranking algorithm using $\mathcal{S}'$ is the same as that obtained by the spectral ranking algorithm using $\mathcal{S}' + \Delta \mathcal{S}'$.*

*Proof:* When only the $i$-th row and $i$-th column of the similar matrix are corrupted, we have $\sum_{t \neq j} \Delta \mathcal{S}'_{jt} = \Delta \mathcal{S}'_{ij}$. Then, based on the proof of Theorem 1,

$$\Delta f_j - \Delta f_i \leq \frac{n-1}{n - 1 - \min_{1 \leq i \leq n}\left\{\sum_{t \neq i} |S'_{it}|\right\}} \left[-\Delta \lambda - |\Delta \mathcal{S}'_{ij}| + \|\Delta \boldsymbol{S}'_{[i,:]}\|_1\right](f_i - f_j) \tag{23}$$

$$\leq f_i - f_j \tag{24}$$

In this case, $\|\Delta \mathcal{S}\|_{\mathbf{F}} = \|\Delta \boldsymbol{S}'_{[i,:]}\|_2$. And the above inequality holds for all $1 \leq j \leq n$, so

$$\|\Delta \boldsymbol{S}'_{[i,:]}\|_2 - \|\Delta \boldsymbol{S}'_{[i,:]}\|_1 + \max_{1 \leq j \leq n} \left|\Delta \mathcal{S}'_{ij}\right| \leq 1 - \frac{\min_{1 \leq i \leq n}\left\{\sum_{t \neq i} |S'_{it}|\right\}}{n-1} \tag{25}$$

**Proof Via Eigenvalue gaps**

**Theorem 3** *For a similarity matrix $\mathcal{S}' \in \mathbb{R}^{n \times n}$, suppose rows $i$ and column $i$ are corrupted due to inadequate feature representations being learnt for sample $i$. When*

$$||E_{[i,:]}||_2 \leq \frac{(\lambda_3 - \lambda_2)\sqrt{min_{j \geq 3} \prod_{z=2, z \neq j}^n |\lambda_z - \lambda_j|}}{3(n-2)\sqrt{\prod_{z=3}^n (\lambda_z - \lambda_2)}},$$

*the seriation obtained by the spectral ranking algorithm using $\mathcal{S}'$ is the same as that obtained by the spectral ranking algorithm using $\mathcal{S}' + E$.*

*Proof:* First, we provide a derivation of the first-order Taylor expansion of Fiedler vector after adding noise. Note that in this case, we only consider that the algebraic multiplicity of the eigenvectors of the Laplacian matrix is all equal to 1.

Let $L'$, $f'$, $\lambda_i'$, $v_i'$, and $L$, $f$, $\lambda_i$, $v_i$ be Laplacian matrix, Fiedler vector, eigenvalue and eigenvector of Laplacian matrix after and before adding noise respectively. $\lambda_1 \leq \lambda_2 \cdots \leq \lambda_n$, $\lambda_1' \leq \lambda_2' \cdots \leq \lambda_n'$.

$f'$ can be written as

$$f' = \alpha \left( f + \sum_{j \neq 2} \beta_j v_j \right), \tag{26}$$

$$\lambda_2' = \lambda_2 + \nu. \tag{27}$$

$\alpha$ is used to ensure that $f'^T f' = 1$. Then,

$$L'f' = \lambda_2' f', \tag{28}$$

$$\Rightarrow (L + L^{(E)})(f + \sum_{j \neq 2} \beta_j v_j) = (\lambda_2 + \nu)(f + \sum_{j \neq 1} \beta_j v_j), \tag{29}$$

$$\Rightarrow L^{(E)}f + \sum_{j \neq 2} \beta_j L^{(E)} v_j = \sum_{j \neq 2} (\lambda_2 - \lambda_j)\beta_j \alpha_j + \nu(f + \sum_{j \neq 2} \beta_j v_j), \tag{30}$$

$$\Rightarrow \nu = f^T L^{(E)} f + \sum_{j \neq 2} \beta_j f^T L^{(E)} v_j, \tag{31}$$

$$\Rightarrow (\lambda_2 - \lambda_j + \nu)\beta_k = v_k^T L^{(E)} f + \sum_{j \neq 2} \beta_j v_k^T L^{(E)} v_j. \tag{32}$$

Equation (19) based on the definition of $f$, equation (20) apply $f^T$ and equation (21) apply $v_k^T$, $k \neq 2$. Expand $\nu, \alpha, \beta_j$ as follows:

$$\nu = 0 + \varepsilon \nu^{(1)} + O(\varepsilon^2), \tag{33}$$

$$\alpha = 1 + \varepsilon \alpha^{(1)} + O(\varepsilon^2), \tag{34}$$

$$\beta_j = 0 + \varepsilon \beta_j^{(1)} + O(\varepsilon^2), \tag{35}$$

where $\varepsilon \to 0$. Let $L^{(E)} = \varepsilon B$ and then combine it with equation (20) and (21), we have

$$\nu^{(1)} = f^T B F, \tag{36}$$

$$\beta_j^{(1)} = \frac{v_k^T B f}{\lambda_2 - \lambda_j}. \tag{37}$$

Next, we prove that $\alpha^{(1)} = 0$.

$$(1 + 2\varepsilon\alpha^{(1)} + O(\varepsilon^2)) \left( 1 + 2\varepsilon \sum_{j \neq 2} \beta_j^{(1)} f^T v_j + O(\varepsilon^2) \right) \tag{38}$$

$$= 1 + 2\varepsilon\alpha^{(1)} + O(\varepsilon^2), \tag{39}$$

$$= 1, \tag{40}$$

$$\Rightarrow \alpha^{(1)} = 0. \tag{41}$$

Therefore,

$$f' = f + \sum_{j \neq 2} \frac{v_j^T L^{(E)} f}{\lambda_2 - \lambda_j} v_j. \tag{42}$$

Let $\Delta f_p = f_p' - f_p, \Delta f_q = f_q' - f_q$. To prove our theorem, it is sufficient to demonstrate that when $f_p \geq f_q, f_p' \geq f_q'$, that is,

$$f_p - f_q \geq \Delta f_q - \Delta f_p, \forall 1 \leq p, q \leq n, \tag{43}$$

where $f_p$ is the $p$-th element of the Fiedler vector $f$ of $\mathcal{S}'$. The following proof is only considering the situation where $f_p, f_q \geq 0, \Delta f_q \leq 0, \Delta f_p \geq 0$. When the given upper bound of the error satisfies the above condition, the other conditions are also satisfied.

According to equation (31), we can derive

$$\Delta f_q - \Delta f_p = \sum_{j \neq 2} \frac{v_j^T L^{(E)} f}{\lambda_2 - \lambda_j} (v_{jq} - v_{jp}). \tag{44}$$

In this section, only the $i$-th row and $i$-th column of the similarity matrix are corrupted, so $E$ and $L^{(E)}$ can be written as follows:

$$E = \begin{bmatrix} & E_{i1} & \\ & \vdots & \\ E_{i1}, \cdots, & 0, & \cdots, E_{in} \\ & \vdots & \\ & E_{in} & \end{bmatrix}.$$

$$L^{(E)} = \begin{bmatrix} & E_{i1} & \\ & E_{i2} & \\ & \vdots & \\ E_{i1}, E_{i2}, \cdots, & \sum_j E_{ij}, \cdots, E_{in} \\ & \vdots & \\ & E_{in} & \end{bmatrix},$$

$$= \begin{bmatrix} 0 & \\ 0 & \\ \vdots & \\ E_{i1}, \cdots, E_{ii}, \cdots, E_{in} \\ \vdots & \\ 0 & \end{bmatrix} + \begin{bmatrix} E_{i1} & \\ E_{i2} & \\ \vdots & \\ 0, \cdots, E_{ii}, \cdots, 0 \\ \vdots & \\ E_{in} & \end{bmatrix} + \begin{bmatrix} & 0 & \\ & 0 & \\ 0, 0, \cdots, & \sum_j E_{ij}, \cdots, 0 \\ & \vdots & \\ & 0 & \end{bmatrix}.$$

It is noted that we assume that the diagonal elements of $\mathcal{S}'$ are not perturbed. This is reasonable since the diagonal elements correspond to cosine similarity between the same features, which will always be 1.

So

$$\sum_{j \neq 2} \frac{v_j^T L^{(E)} f}{\lambda_2 - \lambda_j} (v_{jq} - v_{jp}) = \sum_{j \neq 2} \frac{v_{ji} f^T E_{[i,:]} + f_i v_j^T E_{[i,:]} + f_i v_{ji} e^T E_{[i,:]}}{\lambda_2 - \lambda_j} (v_{jq} - v_{jp}), \quad (45)$$

$$= (\sum_{j \neq 2} \frac{v_{ji} f^T + f_i v_j^T + f_i v_{ji} e^T}{\lambda_2 - \lambda_j} (v_{jq} - v_{jp})) ||E_{[i,:]}||_2, \quad (46)$$

$$\leq \sum_{j \neq 2} \frac{(v_{ji} + f_i + f_i v_{ji})(v_{jp} - v_{jq})}{\lambda_j - \lambda_2} ||E_{[i,:]}||_2. \quad (47)$$

We know that

$$|v_{jp} - v_{jq}| = \sqrt{2 \frac{\prod_{z=2}^{n-1} (\mu_z - \lambda_j)}{\prod_{z=2, z \neq j}^{n} (\lambda_z - \lambda_j)}}, \quad (48)$$

where $0 = \mu_1 < \mu_2 \leq \cdots \leq \mu_{n-1}$ are the eigenvalues of the scaled Laplacian for the $(i, j)$-coalesced graph.

When $||E_{[i,:]}||_2 \leq \frac{(\lambda_3 - \lambda_2) \sqrt{min_{j \geq 3} \prod_{z=2, z \neq j}^{n} |\lambda_z - \lambda_j|}}{3(n-2) \sqrt{\prod_{z=3}^{n} (\lambda_z - \lambda_2)}}$ and combined with equation (36) and equation (37), we can derive that

$$\sum_{j \neq 2} \frac{v_j^T L^{(E)} f}{\lambda_2 - \lambda_j} (v_{jq} - v_{jp}) \leq \sqrt{2 \frac{\prod_{z=2}^{n-1} (\mu_z - \lambda_2)}{\prod_{z=3}^{n} (\lambda_z - \lambda_2)}} = f_p - f_q. \quad (49)$$

$\square$

## S3    Experiments

### S3.1.4    Impact of unlabeled batch size

In our experiments, we use a batch size of 10 samples to perform spectral seriation and enforce supervision on unlabeled samples. We test the sensitivity of our method to the size of the unlabeled batch. We perform this on our synthetic dataset to se how results are affected by different batch sizes for sampling unlabeled data. We show results using batch sizes in $5, 10, 20, 30, 40$ in Table S1. We can see that performance is mostly stable, although a larger batch-size can lead to slightly reduced performance.

Table S1: Results using different batch sizes for unlabeled data sampling

| Batch Size | MAE $\downarrow$ | $R^2 \uparrow$ |
|---|---|---|
| 5 | $0.028 \pm 0.008$ | $96.8\% \pm 1.8$ |
| 10 | $\mathbf{0.027 \pm 0.009}$ | $\mathbf{97.3\% \pm 2.4}$ |
| 20 | $0.028 \pm 0.008$ | $\mathbf{97.3\% \pm 2.2}$ |
| 30 | $0.029 \pm 0.007$ | $97.2\% \pm 1.7$ |
| 40 | $0.033 \pm 0.009$ | $96.4\% \pm 2.3$ |

### S3.1.5    Computational and memory costs

We provide reference times in seconds for performing one iteration of training and inference for different methods in Table S2. Actual times may differ depending on hardware and environment. We note that CLSS does not introduce significant computational complexity since additional calculations involving eigenvalue decomposition can be performed efficiently with existing computational tools and algorithms. Test-time inference is also more efficient than state-of-the-art semi-supervised

Table S2: Computational time in seconds for one iteration of training and inference.

| Type | Method | Training | Testing |
|------|--------|----------|---------|
| *Supervised* | Regression | 0.2015 | 0.0013 |
| | Regression + ODE | 0.2167 | 0.0012 |
| *Semi-supervised* | Mean-teacher | 0.2145 | 0.0012 |
| | CPS | 0.2022 | 0.0018 |
| | UCVME | 0.2487 | 0.0043 |
| | CLSS (Ours) | 0.2310 | 0.0013 |

methods because we only require predictions from one model, instead of taking the average from two co-trained models.

We also show the number of model parameters required for each method in Table S3. We note that CLSS only uses one model, whilst alternative methods rely on two co-trained models which requires doubles the memory.

Table S3: Number of model parameters for each method.

| Type | Method | Number of parameters |
|------|--------|----------------------|
| *Supervised* | Regression | 34,401 |
| | Regression + ODE | 34,401 |
| *Semi-supervised* | Mean-teacher | 68,802 |
| | CPS | 68,802 |
| | UCVME | 69,004 |
| | CLSS (Ours) | 34,401 |

### S3.1.6  Hyper-parameter sensitivity

Hyper-parameters were selected based on a coarse search on the validation set. We show hyper-parameter sensitivity results performed using a quarter of available labels for reference (Table S4). Each parameter is adjusted individually whilst keeping the remaining ones at the optimum value.

Table S4: Results using different hyper-parameter settings

| Hyper-parameter | Value | MAE |
|-----------------|-------|-----|
| $w_{SC}$ | 0.01 | 0.033 |
| | 0.001 | 0.027 |
| | 0.0001 | 0.030 |
| $w_{UC}$ | 0.01 | 0.032 |
| | 0.001 | 0.027 |
| | 0.0001 | 0.031 |
| $w_{UR}$ | 0.01 | 0.052 |
| | 0.001 | 0.027 |
| | 0.0001 | 0.030 |

### S3.2  Validation on Brain Age estimation from MRI Scans

### S3.2.1  Comparison with state-of-the-art alternatives and ablation studies

To analyse the effect of different components in our methodology, namely the use of $\mathcal{L}^{SC}$, $\mathcal{L}^{UC}$, and $\mathcal{L}^{UR}$, we perform training with the loss functions added separately to study their impact. Results are shown in Table S5.

We can see that each individual component leads to contributions in improved performance across all settings. This provides empirical support for our method on a real-world dataset.

Table S5: Ablation study of different components

MAE↓

| Method | SC | UC | UR | 1/5 labels | 1/4 labels | 1/3 labels | 1/2 labels |
|---|---|---|---|---|---|---|---|
| Regression | | | | $9.95 \pm 1.41$ | $11.93 \pm 1.40$ | $11.76 \pm 1.75$ | $10.93 \pm 1.60$ |
| Regression+$\mathcal{L}^{SC}$ | ✓ | | | $10.55 \pm 1.94$ | $10.61 \pm 1.64$ | $11.21 \pm 1.90$ | $11.62 \pm 1.50$ |
| Regression+$\mathcal{L}^{SC}$+$\mathcal{L}^{UC}$ | ✓ | ✓ | | $9.97 \pm 1.54$ | $10.04 \pm 1.56$ | $9.83 \pm 1.56$ | $9.43 \pm 1.57$ |
| Ours | ✓ | ✓ | ✓ | $\mathbf{9.58 \pm 1.48}$ | $\mathbf{9.68 \pm 1.22}$ | $\mathbf{9.72 \pm 1.29}$ | $\mathbf{9.37 \pm 1.17}$ |

$R^2$↑

| Method | ODE | ULB | PSL | 1/5 labels | 1/4 labels | 1/3 labels | 1/2 labels |
|---|---|---|---|---|---|---|---|
| Regression | | | | $43.1\% \pm 14.4$ | $20.2\% \pm 16.3$ | $23.3\% \pm 20.7$ | $33.3\% \pm 16.7$ |
| Regression+ODE | ✓ | | | $36.1\% \pm 20.1$ | $33.5\% \pm 18.2$ | $30.0\% \pm 20.9$ | $24.8\% \pm 17.5$ |
| Regression+ODE+ULB | ✓ | ✓ | | $41.9\% \pm 15.6$ | $40.6\% \pm 17.0$ | $44.6\% \pm 16.3$ | $47.0\% \pm 16.7$ |
| Ours | ✓ | ✓ | ✓ | $\mathbf{45.0\% \pm 17.6}$ | $\mathbf{44.5\% \pm 11.5}$ | $\mathbf{44.9\% \pm 14.9}$ | $\mathbf{48.9\% \pm 13.2}$ |

### S3.2.2 Hyper-parameter sensitivity

Hyper-parameters were selected based on a coarse search on the validation set. We show hyper-parameter sensitivity results performed using half of available labels for reference (Table S6). Each parameter is adjusted individually whilst keeping the remaining ones at the optimum value.

Table S6: Results using different hyper-parameter settings

| Hyper-parameter | Value | MAE |
|---|---|---|
| | 5 | 12.73 |
| $w_{SC}$ | 1 | 9.37 |
| | 0.2 | 9.42 |
| | 0.5 | 10.24 |
| $w_{UC}$ | 0.05 | 9.37 |
| | 0.005 | 9.45 |
| | 0.1 | 9.49 |
| $w_{UR}$ | 0.01 | 9.37 |
| | 0.001 | 10.61 |

### S3.3 Validation on Age-Estimation from photographs

### S3.3.1 Comparison with state-of-the-art alternatives and ablation studies

We add the loss values $\mathcal{L}^{SC}$, $\mathcal{L}^{UC}$, and $\mathcal{L}^{UR}$ separately during training to investigate their individual impact for age estimation from photographs. Results are shown in Table S7.

Table S7: Ablation study of different components

MAE↓

| Method | SC | UC | UR | 1/30 labels | 1/25 labels | 1/20 labels | 1/15 labels |
|---|---|---|---|---|---|---|---|
| Regression | | | | $10.14 \pm 0.25$ | $9.99 \pm 0.11$ | $9.10 \pm 0.15$ | $8.58 \pm 0.10$ |
| Regression+$\mathcal{L}^{SC}$ | ✓ | | | $10.02 \pm 0.23$ | $9.87 \pm 0.20$ | $8.97 \pm 0.14$ | $8.51 \pm 0.12$ |
| Regression+$\mathcal{L}^{SC}$+$\mathcal{L}^{UC}$ | ✓ | ✓ | | $9.97 \pm 0.18$ | $9.62 \pm 0.14$ | $\mathbf{8.88 \pm 0.12}$ | $8.49 \pm 0.10$ |
| Ours | ✓ | ✓ | ✓ | $\mathbf{9.95 \pm 0.18}$ | $\mathbf{9.59 \pm 0.12}$ | $8.89 \pm 0.09$ | $\mathbf{8.45 \pm 0.11}$ |

$R^2$↑

| Method | SC | UC | UR | 1/30 labels | 1/25 labels | 1/20 labels | 1/15 labels |
|---|---|---|---|---|---|---|---|
| Regression | | | | $63.6\% \pm 1.6$ | $64.5\% \pm 0.8$ | $70.4\% \pm 0.9$ | $72.9\% \pm 0.8$ |
| Regression+$\mathcal{L}^{SC}$ | ✓ | | | $63.9\% \pm 1.5$ | $65.4\% \pm 1.3$ | $70.8\% \pm 0.9$ | $73.2\% \pm 0.6$ |
| Regression+$\mathcal{L}^{SC}$+$\mathcal{L}^{UC}$ | ✓ | ✓ | | $64.1\% \pm 1.0$ | $66.6\% \pm 0.7$ | $\mathbf{71.5\% \pm 0.7}$ | $73.5\% \pm 0.6$ |
| Ours | ✓ | ✓ | ✓ | $\mathbf{64.5\% \pm 1.0}$ | $\mathbf{66.9\% \pm 0.7}$ | $71.3\% \pm 0.7$ | $\mathbf{73.7\% \pm 0.6}$ |

We can see that $\mathcal{L}^{SC}$ and $\mathcal{L}^{UC}$ both lead to significant improvements in performance, demonstrating the effectiveness of using spectral seriation for contrastive learning on unsupervised samples. Further improvements from using $\mathcal{L}^{UR}$ are more limited.

### S3.3.2 Hyper-parameter sensitivity

Hyper-parameters were selected based on a coarse search on the validation set. We show hyper-parameter sensitivity results performed using 1/25 of available labels for reference (Table S8). Each parameter is adjusted individually whilst keeping the remaining ones at the optimum value.

Table S8: Results using different hyper-parameter settings

| Hyper-parameter | Value | MAE |
|---|---|---|
| $w_{SC}$ | 5 | 9.73 |
| | 1 | 9.59 |
| | 0.2 | 9.91 |
| $w_{UC}$ | 0.5 | 10.70 |
| | 0.05 | 9.59 |
| | 0.005 | 9.64 |
| $w_{UR}$ | 0.1 | 9.97 |
| | 0.01 | 9.59 |
| | 0.001 | 9.65 |