# OpenReview forum: "Semi-Supervised Contrastive Learning for Deep Regression with Ordinal Rankings from Spectral Seriation"
_NeurIPS.cc/2023/Conference — NeurIPS 2023 poster_

### Official Review · Reviewer_w9AQ · 2023-06-23

**Soundness:** 3 good
**Presentation:** 2 fair
**Contribution:** 3 good
**Rating:** 5
**Confidence:** 3

**Summary:**

This paper tried to deal with semi-supervised deep regression problems via contrastive learning. To fully use the unlabeled data, they let the feature similarity between unlabeled samples be in agreement with the ranks. Therefore, the accurate ordinal relationship can be recovered through spectral seriation algorithms via feature similarity.

**Strengths:**

1. The paper uses contrastive learning for unlabeled data by fully considering the ordinal relationship.
2. The paper adopted a robust ranking method named spectral seriation and boost the overall contrastive performance.
3.  Real-world application experiments on medical imaging processing demonstrate its effectiveness and significance.

**Weaknesses:**

1. It seems that the paper does not have obvious weaknesses, if some, the main weaknesses are in the model design Section. I do not understand how to optimize equation (7).

2. Some descriptions need to be more clear. For example:

    Line 3, "only unlike for" -> "only, unlike for"?

    Line 28 said,  "This is not possible...". What does "This" refer to?

   Line 105, what is "Fiedler vector"?

3. Some typos:

    Line 105 "ofL".
    Equation (7), what is the meaning of  $R^{'} - R^{'}_{[i]}$?


**Questions:**

1. Why does the proposed method need to rank the supervision of unlabeled predictions with seriation rankings?
2. Have you considered other baseline methods? Like a simple SimClr  method for unlabeled data?
3. Equation (7), what is the meaning of  $R^{'} - R^{'}_{[i]}$? Could you provide more explanations?



**Limitations:**

Yes

---

> ### Author Rebuttal · Authors · 2023-08-04
>
> Thank you for the positive comments! We address remaining concerns below.
> \
> &nbsp;
>
> ### W1) Optimizing Eq. 7
>
>
> We make use of the differentiable combinatorial solver proposed in [1] to optimize Eq. 7. The loss function $\ell$ enforces predictions for discrete combinatorial values, such as rankings, to follow some ground-truth order. A high level explanation of $\ell$ is that it makes use of interpolation methods to allow the discrete combinatorial predictions to be differentiable. We do not provide detailed descriptions of this function since we directly use the implementation from [1], and it is better explained in the original work. We will include an intuitive explanation in our revision however.
> \
> &nbsp;
>
> ### W2) Clearer descriptions
>
> >Line 28 said, "This is not possible...". What does "This" refer to?
>
> "This" refers to using unlabeled data only for unsupervised contrastive learning.
>
> >Line 105, what is "Fiedler vector"?
>
> The Fiedler vector is defined as the eigenvector corresponding to the smallest non-zero eigenvalue of a matrix [2].
>
>
> We will make the necessary clarifications in the revision.
> \
> &nbsp;
>
> ### W3) Meaning of $R' - R'_{[i]}$
>
> $R' - R'_{[i]}$ is a vector obtained by subtracting every element in the ranking vector $R'$ by the $i$th element of $R'$.
>
>
> For example, if
>
>
> $R' = [1,5,2,3,4]$,
>
>
> $R' - R'_{[3]}$ is calculated by subtracting every element in $R'$ by the third entry of $R'$, i.e. 2.
>
>
> This gives us $[-1,3,0,1,2]$.
>
>
> This is used to supervise the relative relationships between similarity values in matrix $S'$. Intuitively, if the ranking of sample $i$ is closer to sample $j$ compared to sample $k$, then the cosine similarity between samples ${i,j}$ should also be greater than samples ${i,k}$.  $R' - R'_{[i]}$ calculates the ranking difference between samples relative to sample $i$.
> \
> &nbsp;
>
> ### Q1) Supervising unlabeled predictions with seriation rankings
>
>
> We note that the rankings obtained through spectral seriation is robust to noise and has error correction properties. Theoretical bounds are derived in Theorems 2 and 3.
>
>
> Because of this, using the recovered rankings to supervise output predictions should have additional benefits compared to without supervision. Furthermore, this also ensures that the feature relationships learnt through contrastive learning are consistent with the predicted outputs on unlabeled samples.
>
>
> We empirically verify that including this constraint leads to additional improvements through ablation experiments in Figure 3 and Tables S2 and S3 of the supplementary materials.
> \
> &nbsp;
>
> ### Q2) Using SimCLR for unlabeled samples
>
> Contrastive learning methods for classification generally do not work well when applied to regression, since they are unable to consider label distance and order relationships. The loss function treats all negative pairs equivalently without regard to its label value, which often results in ineffective features being learnt. This has been verified in existing works such as [3-4].
>
> We can also confirm that applying SimCLR to unlabeled samples only also does not work well, as it disrupts features learnt by the contrastive regression loss function on labeled samples. We show results when applied to our three tasks.
>
>
> **Synthetic dataset, 1/4 labels (Section 3.1)**
> | |MAE|
> |---|---|
> |SimCLR on unlabeled samples|0.059|
> |CLSS (ours) |0.027|
>
> **Brain age estimation, 1/2 labels (Section 3.2)**
> | |MAE|
> |---|---|
> |SimCLR on unlabeled samples|10.95|
> |CLSS (ours) |9.37|
>
>
> **Age estimation, 1/25 labels (Section 3.3)**
>
>
> | |MAE|
> |---|---|
> |SimCLR on unlabeled samples|9.91|
> |CLSS (ours) |9.59|
>
>
> \
> &nbsp;
>
> ### Q3) Meaning of $R' - R'_{[i]}$
>
> See response to W3.
> \
> &nbsp;
>
>
> [1] Pogančić, Marin Vlastelica, et al. "Differentiation of blackbox combinatorial solvers." ICLR. 2019.
>
> [2] Atkins, Jonathan E., Erik G. Boman, and Bruce Hendrickson. "A spectral algorithm for seriation and the consecutive ones problem." SIAM Journal on Computing 28.1 (1998): 297-310.
>
> [3] Dai, Weihang, et al. "Adaptive contrast for image regression in computer-aided disease assessment." IEEE Transactions on Medical Imaging 41.5 (2021): 1255-1268.
>
> [4] Zhang, Shihao, et al. "Improving Deep Regression with Ordinal Entropy." arXiv preprint arXiv:2301.08915 (2023).

---

> > ### Comment · Reviewer_w9AQ · 2023-08-16
> > **After the rebuttal**
> >
> > Thanks for the responses. Most of my concerns have been addressed, I am happy to keep my score.

---

### Official Review · Reviewer_kMK1 · 2023-07-04

**Soundness:** 3 good
**Presentation:** 3 good
**Contribution:** 3 good
**Rating:** 5
**Confidence:** 3

**Summary:**

 This paper extends contrastive regression methods to a semi-supervised setting using unlabeled data. They leverage the feature similarity matrix between these samples to infer ordinal relationships, a process guaranteed robust if the error is within defined bounds.

**Strengths:**

This paper successfully extends contrastive regression to include unlabeled datasets, marking a significant step in semi-supervised learning. The authors' writing style is commendable clear, succinct, and easy to follow. Notably, the proposed method isn't just empirical; it is firmly grounded in theoretical evidence, which enhances its credibility and potential for practical application.


**Weaknesses:**

* Ablation studies are missing, ref L2,3
* Missing explanations on loss function design, for each component, what the purpose
* Section 2.3 is not clear, ref Q5



**Questions:**

* Why "contrastive learning that are only able to use labeled data" ? contrastive learning, e,g, Simclr, MOCO, etc are under unlabeled data
* How to achieve similarity matrix S?
* why did you set the loss function like formulation (9), for each part, would you give intuitive explanations?
* In eq (7,8,9), how 's the ranking function $l()$ look like?
* For section 2.3, what are the intuitive explanations of Theorem 2,3, what insights can be taken away? what do the theorems trying to prove?

**Limitations:**

* Ablation studies on each loss component are missing
* Ablation studies on hyperparameters are missing
* Section 2.3 is not clear enough


I may have misunderstood, if you solve my question, I am willing to increase the score

---

> ### Author Rebuttal · Authors · 2023-08-04
>
> Thank you for the encouraging feedback! We address questions below.
> \
> &nbsp;
>
> ### Q1) Labels for contrastive learning
>
>
> To clarify, it is possible to perform contrastive learning using unlabeled data **for classification** (e.g. SimCLR, MOCO).
>
>
> It is **NOT possible** for existing contrastive learning methods to use unlabeled data **for regression**.
>
>
> This is because regression labels **reflect order and distance relationships between samples** [1-2], but for classification, we only need to distinguish between different classes. Effective feature representations for regression therefore **need to reflect label distance relationships** unlike for classification.
>
> Intuitively, for samples $i,j,k$, if we have $y_i=5$, $y_j=6$, and $y_k=20$, we expect features for sample $i$ should be more similar to sample $j$ compared to sample $k$ since their labels are closer together. To enforce this, **it is necessary to know the labels**.
>
> In our work, we propose a novel method that **extracts order relationships from unlabeled samples as well** through spectral seriation for contrastive learning.
> \
> &nbsp;
>
> ### Q2) Calculating similarity matrix $S$
>
>
> $S$ is obtained by calculating cosine similarity between features within a batch.
>
>
> We use $\tilde{z}$ to denote L2 normalized features. $\tilde{z}$ has dimensions $B \times F$ where $B$ is the batch size and $F$ is the feature length.
>
> $S$ is equal to $\tilde{z} \tilde{z}^T$ and has dimensions $B \times B$.
> \
> &nbsp;
>
> ### Q3) Explanation of loss function (Eq. 9)
>
>
> Eq. 9: $\mathcal{L} = \mathcal{L}^{SR} + w_{SC} \mathcal{L}^{SC}  + w_{UC} \mathcal{L}^{UC} + w_{UR} \mathcal{L}^{UR}$
>
>
> $\mathcal{L}^{SR}$ is the supervised regression loss (Eq. 2).
>
>
> $\mathcal{L}^{SC}$ is the supervised contrastive loss for regression. We use Ordinal Entropy [2] in our experiments (line 184). This ensures that the model learns features that are consistent with the distance relationships between sample labels. Pairs with labels closer together will have features that are more similar compared to pairs with labels further apart.
>
>
> $\mathcal{L}^{UC}$ is the unsupervised contrastive loss for unlabeled samples (Eq. 7). Because labeled features are trained using contrastive learning, the similarity matrix for unlabeled features will also reflect order and distance relationships between them, but with noise. We can use spectral seriation to extract the ordinal ranking between unlabeled samples from this noisy similarity matrix.
>
>
> By obtaining the ordinal ranking of unlabeled samples, we can infer the relative relationship of similarity values for feature pairs. Intuitively, if sample $i$ is ranked closer to sample $j$ than to sample $k$, then the cosine similarity between feature pairs ${i,j}$ should be larger then ${i,k}$. These relationships are enforced through $\mathcal{L}^{UC}$, thus allowing us to perform contrastive learning using unlabeled samples.
>
>
> $\mathcal{L}^{UR}$ is the unsupervised ranking loss, which we impose on predictions of unlabeled samples (Eq. 8). Because spectral seriation is robust to noise, the recovered ordinal rankings should also be useful for supervising prediction outputs. This also enforces consistency in the order of features and predictions. We enforce the same pairwise relationships inferred in $\mathcal{L}^{UC}$ on predictions using $\mathcal{L}^{UR}$.
> \
> &nbsp;
>
> ### Q4) Ranking function $\ell$ in Eq. 7-9
>
>
> We directly use the differential combinatorial solver proposed in [3] for $\ell$. On a high level, $\ell$ uses interpolation methods to allow discrete combinatorial inputs (e.g. rankings), to be differentiable. We do not provide details since it is better explained in the original work. However, we will include an intuitive explanation in our revision.
> \
> &nbsp;
>
> ### Q5) Explanation of Section 2.3
>
> One advantage of spectral seriation is that it is based on error minimization and is therefore robust to noise. To understand this intuitively, we can imagine a loss surface and its optimum point at the surface minimum. The optimum point will remain the same even if the surface is perturbed, as long as the distortions are within certain limits.
>
>
> Theorems 2 and 3 illustrates this formally through deriving theoretical bounds for noise tolerance. We demonstrate robustness to two different kinds of noise: noise randomly distributed across the entire surface (Theorem 2), and noise from a single poorly learned feature (Theorem 3).
> \
> &nbsp;
>
> ### L1) Loss component ablation
>
>
> Component ablation results on the synthetic dataset are illustrated in Figure 3. Results for brain age estimation and age estimation from photos are given in supplementary materials Table S2 and S3.
> \
> &nbsp;
>
> ### L2) Hyperparameter ablation
>
> We perform hyperparameter ablation by adjusting parameters individually and keeping others at optimum:
>
>
> **Synthetic dataset, 1/4 labels (Section 3.1)**
>
>
> Optimum at $w_{SC}=w_{UC}=w_{UR}=0.001$; MAE 0.027
> | |MAE|
> |---|---|
> |$w_{SC}=0.01$|0.033|
> |$w_{SC}=0.0001$|0.030|
> |$w_{UC}=0.01$|0.032|
> |$w_{UC}=0.0001$|0.031|
> |$w_{UR}=0.01$|0.052|
> |$w_{UR}=0.0001$|0.030|
>
>
> **Brain age estimation, 1/2 labels (Section 3.2)**
>
>
> Optimum at $w_{SC}=1,w_{UC}=0.05,w_{UR}=0.01$; MAE 9.37
> | |MAE|
> |---|---|
> |$w_{SC}=5$|12.73|
> |$w_{SC}=0.2$|9.42|
> |$w_{UC}=0.5$|10.24|
> |$w_{UC}=0.005$|9.45|
> |$w_{UR}=0.1$|9.49|
> |$w_{UR}=0.001$|10.61|
>
>
> **Age estimation, 1/25 labels (Section 3.3)**
>
>
> Optimum at $w_{SC}=1,w_{UC}=0.05,w_{UR}=0.01$; MAE 9.59
> | |MAE|
> |---|---|
> |$w_{SC}=5$|9.73|
> |$w_{SC}=0.2$|9.91|
> |$w_{UC}=0.5$|10.70|
> |$w_{UC}=0.005$|9.64|
> |$w_{UR}=0.1$|9.97|
> |$w_{UR}=0.001$|9.65|
>
> ### L3)
> See Q5.
> \
> &nbsp;
>
> [1] W, Dai et al. "Adaptive contrast for image regression in computer-aided disease assessment." IEEE Transactions on Medical Imaging 41.5 (2021): 1255-1268.
>
> [2] S. Zhang et al. "Improving Deep Regression with Ordinal Entropy." arXiv preprint arXiv:2301.08915 (2023).
>
> [3] M.V. Pogančić, et al. "Differentiation of blackbox combinatorial solvers." ICLR. 2019.

---

> > ### Comment · Reviewer_kMK1 · 2023-08-18
> > **Reply to author**
> >
> > Thank you for providing the answers, which clear up some of my confusion. So, I keep my score.

---

### Official Review · Reviewer_H1Yh · 2023-07-05

**Soundness:** 3 good
**Presentation:** 2 fair
**Contribution:** 3 good
**Rating:** 7
**Confidence:** 4

**Summary:**

In this paper, the authors propose a sophisticated method for deep semi-supervised learning using contrastive loss function. The main idea is to estimate the ordinal relations among the unlabeled data samples by using the similarity matrix of the unlabeled data samples and then use these relations to improve the contrastive learning. To this end, the authors utilize the spectral seriation algorithm of [2]. The proposed method is tested on several datasets and the authors report better accuracies compared to the state-of-the-art.

**Strengths:**

The main strengths of the paper can be summarized as follows:
1) The paper is generally well-written with the exception of Related Work section.
2) The proposed method is sound and supported with theoretical arguments. Estimating the ordinal rankings of the unlabeled data via similarity matrix makes sense and it seems the idea is working. I really liked the idea.
3) The proposed method achieves the state-of-the-art accuracies on all tested datasets.


**Weaknesses:**

The main limitations of the paper can be summarized as follows:
1) Related work section is quite weak and its location in the manuscript is wrong. Please move it to Section 1 after Introduction. Also, there are some important related missing references. I listed some of them below. Especially, the one using graphs and Laplacian matrix is directly related to the proposed method is here since the ordinal rankings are found by using Laplacian matrix in the paper.
2) Using cosine similarities only is a big limitation since it constraints all the feature space onto the boundary of a hypersphere. Please note that two samples very far from each other in the feature space may collapse to the similar points on the unit hypersphere. In contrastive loss, Euclidean distances are more preferred unless additional tricks are not done in the regression head (e.g., ArcFace, Cosface etc. normalize both features and classifier weights in their classification losses).
3) I wonder if the authors use the loss function given in (9) directly. I do not see any constraint to enforce the feature samples to lie on the boundary of the unit hypersphere. In that case, using cosine distances is not a good choice.
4) Please explain how the Fiedlar vector is computed.
5) There are 3 parameters in the proposed loss function. Please describe how they are set.
References
[R1] Mohan Timilsina,  Alejandro Figueroa, Mathieu d’Aquin, Haixuan Yang. Semi-supervised regression using diffusion on graphs, Applied Soft Computing Journal, 2021.
[R2] Jean et al., Semi-supervised Deep Kernel Learning: Regression with Unlabeled Data by Minimizing
Predictive Variance, Neurips, 2018.
[R3] Xu et al., Semi-supervised regression with manifold: A Bayesian deep kernel learning approach, Neurocomputing, 2022.
[R4] Fazakis et al., A multi-scheme semi-supervised regression approach, Pattern Recognition Letters, 2019.


**Questions:**

1) I wonder if the authors use the loss function given in (9) directly. I do not see any constraint to enforce the feature samples to lie on the boundary of the unit hypersphere. In that case, using cosine distances is not a good choice.
2) Please explain how the Fiedlar vector is computed.


**Limitations:**

The authors do not address any limitation, but the main limitation is using cosine distances in the proposed method.

---

> ### Author Rebuttal · Authors · 2023-08-03
>
> Thank you for your detailed comments! We are glad you liked our idea! We address remaining concerns below:
> \
> &nbsp;
>
> ### W1) Improvements to Related Work section
>
> Thank you for your helpful feedback. These will be considered in our revision.
> \
> &nbsp;
>
> ### W2) Use of cosine similarity
>
> It is true that using cosine similarity limits features to a unit hypersphere after normalization. However, existing state-of-the-art contrastive learning methods for classification (e.g. SimCLR [1], MOCO [2], SupCon [3], etc.) and regression (AdaCon [4], Ordinal Entropy [5]), all make use of L2 normalized features and cosine similarity for contrastive learning.
>
> Because the focus of this work is on **how to utilize additional signals from unlabeled samples** for semi-supervised contrastive regression, we build on top of these existing contrastive regression approaches. We do not explore improved contrastive learning loss functions that do not make use of cosine similarity in this work.
>
> Specifically, our method (CLSS) utilizes Ordinal Entropy [5] for contrastive loss, which is implemented after applying L2 normalization on the feature layer (see lines 73, 75, 80, and 184). Regression predictions are obtained using a fully-connected regression layer directly after the features without L2 normalization.
> \
> &nbsp;
>
> ### W3) Loss function in Eq. 9
>
> The loss function in Eq. 9 is applied directly. To clarify, we do not **explicitly enforce** the features to lie on a unit hypersphere. Instead, we apply contrastive loss to the feature layer **after first applying L2 normalization**. This is also the standard approach used in existing state-of-the-art methods for contrastive learning [1-5]. In our method, we use Ordinal Entropy [5] for contrastive learning, which applies L2 normalization on the feature layer before applying the loss function (see lines 73, 75, 80, and 184). The fully-connected regression layer is directly applied to features without normalization to obtain the regression prediction. We will state this more clearly in our revision.
> \
> &nbsp;
>
> ### W4) Computation of Fiedlar vector
>
> The Fiedlar vector is defined as the eigenvector corresponding to the smallest non-zero eigenvalue of a matrix [6]. It can be computed through standard eigenvalue decomposition. We will state this more clearly in our revision.
> \
> &nbsp;
>
> ### W5) Parameters for loss function
>
> The three parameters for the loss function in Eq. 9 ($w_{SC}, w_{UC}, w_{UR}$) are determined by performing a coarse search on the validation set. We include hyperparameter ablation results below for reference. Parameters are individually adjusted whilst keeping the remaining ones at optimum value:
>
>
> **Synthetic dataset, 1/4 labels (Section 3.1)**
>
>
> Optimum at $w_{SC}=w_{UC}=w_{UR}=0.001$; MAE 0.027
> | |MAE|
> |---|---|
> |$w_{SC}=0.01$|0.033|
> |$w_{SC}=0.0001$|0.030|
> |$w_{UC}=0.01$|0.032|
> |$w_{UC}=0.0001$|0.031|
> |$w_{UR}=0.01$|0.052|
> |$w_{UR}=0.0001$|0.030|
>
>
> **Brain age estimation, 1/2 labels (Section 3.2)**
>
>
> Optimum at $w_{SC}=1,w_{UC}=0.05,w_{UR}=0.01$; MAE 9.37
> | |MAE|
> |---|---|
> |$w_{SC}=5$|12.73|
> |$w_{SC}=0.2$|9.42|
> |$w_{UC}=0.5$|10.24|
> |$w_{UC}=0.005$|9.45|
> |$w_{UR}=0.1$|9.49|
> |$w_{UR}=0.001$|10.61|
>
>
> **Age estimation, 1/25 labels (Section 3.3)**
>
>
> Optimum at $w_{SC}=1,w_{UC}=0.05,w_{UR}=0.01$; MAE 9.59
> | |MAE|
> |---|---|
> |$w_{SC}=5$|9.73|
> |$w_{SC}=0.2$|9.91|
> |$w_{UC}=0.5$|10.70|
> |$w_{UC}=0.005$|9.64|
> |$w_{UR}=0.1$|9.97|
> |$w_{UR}=0.001$|9.65|
>
> \
> &nbsp;
> \
> &nbsp;
>
> ### Q1) Loss function in Eq. 9
>
> See response to W3.
> \
> &nbsp;
>
> ### Q2) Computation of Fiedlar vector
>
> See response to W4.
> \
> &nbsp;
>
> [1] Chen, Ting, et al. "A simple framework for contrastive learning of visual representations." International conference on machine learning. PMLR, 2020.
>
>
> [2] He, Kaiming, et al. "Momentum contrast for unsupervised visual representation learning." Proceedings of the IEEE/CVF conference on computer vision and pattern recognition. 2020.
>
>
> [3] Khosla, Prannay, et al. "Supervised contrastive learning." Advances in neural information processing systems 33 (2020): 18661-18673.
>
>
> [4] Dai, Weihang, et al. "Adaptive contrast for image regression in computer-aided disease assessment." IEEE Transactions on Medical Imaging 41.5 (2021): 1255-1268.
>
>
> [5] Zhang, Shihao, et al. "Improving Deep Regression with Ordinal Entropy." arXiv preprint arXiv:2301.08915 (2023).
>
>
> [6] Atkins, Jonathan E., Erik G. Boman, and Bruce Hendrickson. "A spectral algorithm for seriation and the consecutive ones problem." SIAM Journal on Computing 28.1 (1998): 297-310.

---

> > ### Comment · Reviewer_H1Yh · 2023-08-18
> > **reply to the author**
> >
> > The authors should seriously consider to move the related work subsection to the Introduction. Using L2 norm before the loss already enforces the samples to lie on the boundary of the hyperpshere and this is a necessary step if one uses cosine distances, this addresses teh issue I pointed out. In general, I liked the paper and I keep my initial rating, accept.

---

### Official Review · Reviewer_jwFg · 2023-07-06

**Soundness:** 3 good
**Presentation:** 3 good
**Contribution:** 3 good
**Rating:** 6
**Confidence:** 3

**Summary:**

The paper presents a novel approach towards extending contrastive learning methods for deep regression in a semi-supervised setting. The authors address the challenge of using unlabeled data for contrastive learning in deep regression tasks by applying spectral seriation algorithms to infer the ordinal relationship between unlabeled samples. Empirically, they show that the proposed method improves performance on medical datasets.

**Strengths:**

- The motivation for the paper is clear, and the idea is novel from what I can tell.
- The authors provide a comprehensive robustness analysis of their method, with theoretical proofs showing its resilience to different types of noise.
- The paper is well-organized and logically structured, with clear explanations.

**Weaknesses:**

I understand that the motivation for this paper is mainly for medical datasets, but it would be great to test whether the proposed method also works beyond medical setting, as it does seem quite general.

**Questions:**

- What is the computational complexity of the proposed method? How does the training and inference time compare to other state-of-the-art methods?
 - How well would the proposed method perform on more diverse and complex datasets, especially beyond the medical setting?

---

> ### Author Rebuttal · Authors · 2023-08-03
>
> Thank you for your positive comments! We are glad you found our idea novel and well supported theoretically. We address remaining questions below:
> \
> &nbsp;
>
> ### W1) Application beyond medical settings
>
> See response to Q2
> \
> &nbsp;
>
> ### Q1) Computational complexity
>
> Our proposed method, CLSS, does not introduce significant computational complexity compared to state-of-the-art methods. CLSS requires calculating eigenvalues and eigenvectors for Laplacian matrix $L$ (Eq. 6), but this can be done efficiently with existing computational tools and algorithms. Test-time inference is also more efficient than state-of-the-art semi-supervised methods because we only require predictions from **one** model, instead of taking the average from **two** co-trained models.
>
> We show some reference times below for the synthetic dataset used in Section 3.1. We report the time taken in seconds for one iteration of training and inference for different methods. We use the same batch sizes for all methods as stated in the manuscript for fair comparison.
>
>
> | |Training (seconds per iteration)|Testing (seconds per iteration)|
> | ----------- | ----------- | ----------- |
> | Regression | 0.2015 | 0.0013 |
> | Regression + Contrastive Loss | 0.2167 | 0.0012 |
> | Mean-teacher | 0.2145 | 0.0012 |
> | CPS | 0.2022 | 0.0018 |
> | UCVME| 0.2487 | 0.0043 |
> | CLSS (Ours)| 0.2310| 0.0013|
>
>
> Our method CLSS has competitive inference times for testing (0.0013). Training time is also faster than UCVME (0.2310 vs 0.2487), the best performing alternative, since UCVME performs variational inference during training.
>
> CLSS also has smaller model size compared to state-of-the-art semi-supervised approaches. CLSS only uses **one** model, whilst alternative methods rely on **two** co-trained models, thereby doubling the memory required.
>
> | |Number of parameters|
> | ----------- | ----------- |
> | Regression | 34,401 |
> | Regression + Contrastive Loss | 34,401 |
> | Mean-teacher | 68,802 |
> | CPS | 68,802 |
> | UCVME| 69,004 |
> | CLSS (Ours)| 34,401|
>
> Overall, our method is computationally efficient and memory efficient. We will include these details in our revision.
> \
> &nbsp;
> ### Q2) Performance on general datasets
>
> To clarify, we performed experiments on **three different types of datasets** in our manuscript:
> 1. a synthetic dataset for solving partial differential equations (PDEs) through operator learning (Section 3.1)
> 2. a medical dataset for brain age estimation from MRI scans (Section 3.2)
> 3. a natural image dataset for age estimation from photographs (Section 3.3)
>
> This is also stated at the start of Section 3.
>
> In the manuscript, we emphasize the benefits of our method for medical imaging analysis because semi-supervised deep regression is particularly valuable for such realistic problem settings. However, our experiments demonstrate that CLSS **can similarly be applied to different problem types**, such solving PDEs and regression benchmark tasks like age estimation.
>
> We will make sure that this is stated more explicitly in our revision to avoid confusion.

---

### Author Rebuttal · Authors · 2023-08-08

We thank the reviewers for taking the time to provide their thoughtful comments and feedback. We are glad that in general, the reviewers found our method novel and well supported theoretically. We individually address remaining questions and concerns below. These will also be included in our final revision to help strengthen the manuscript.

---

### Decision · Program_Chairs · 2023-09-21

**Decision:**

Accept (poster)

**Comment:**

Summary: A solid paper that addresses contrastive learning for regression that self-supervises by enforcing rankings of similarity to align with the direct rankings of the cosine similarity and predicted y-labels. The reason why the former helps is detailed in Section 2.3. which shows how seriation can denoise noisy similarity matrices.

The paper is well-written. Some of the experimental results do suggest that the method beats baselines, though many lie in the error bars of previous methods. The rebuttal was on point and addressed the reviewers' questions well.